# Safety Evaluation for Acute and Chronic Oral Toxicity of Maha Pigut Triphala Contains Three Medicinal Fruits in Sprague-Dawley Rats

**DOI:** 10.3390/biology13121005

**Published:** 2024-12-02

**Authors:** Supaporn Intatham, Weerakit Taychaworaditsakul, Parirat Khonsung, Sunee Chansakaow, Kanjana Jaijoy, Nirush Lertprasertsuke, Noppamas Soonthornchareonnon, Seewaboon Sireeratawong

**Affiliations:** 1Clinical Research Center for Food and Herbal Product Trials and Development (CR-FAH), Faculty of Medicine, Chiang Mai University, Chiang Mai 50200, Thailand; intatham_s@outlook.com; 2Department of Pharmacology, Faculty of Medicine, Chiang Mai University, Chiang Mai 50200, Thailand; wparirat@hotmail.com; 3Department of Biochemistry, Faculty of Medicine, Chiang Mai University, Chiang Mai 50200, Thailand; weerakit.tay@cmu.ac.th; 4Department of Pharmaceutical Sciences, Faculty of Pharmacy, Chiang Mai University, Chiang Mai 50200, Thailand; sunee.c@cmu.ac.th; 5McCormick Faculty of Nursing, Payap University, Chiang Mai 50000, Thailand; joi.kanjana@gmail.com; 6Department of Pathology, Faculty of Medicine, Chiang Mai University, Chiang Mai 50200, Thailand; nlertpra@hotmail.com; 7Department of Pharmacognosy, Faculty of Pharmacy, Mahidol University, Bangkok 10400, Thailand; noppamas.sup@mahidol.ac.th

**Keywords:** Maha Pigut Triphala, *Terminalia bellirica*, *Terminalia chebula*, *Emblica officinalis*, *Phyllanthus emblica*, acute oral toxicity, chronic oral toxicity, safety, Sprague-Dawley rats

## Abstract

Maha Pigut Triphala is a herbal formula composed of dry fruits of three plants, namely, *Terminalia bellirica*, *Terminalia chebula*, and *Emblica officinalis* also known as *Phyllanthus emblica*, which are used in different proportions in traditional Thai medicine. This formula has long been used to treat colds, relieve coughing, and alleviate phlegm. Nevertheless, there have been no studies on the safety of Maha Pigut Triphala in the ratio of 2:1:3. Thereby, this study aimed to investigate the acute and chronic oral toxicity of Maha Pigut Triphala in the ratio of 2:1:3 in Sprague-Dawley rats. The results showed that the rats that were given a single dose of Maha Pigut Triphala and those that received Maha Pigut Triphala once daily for 270 days in acute and chronic oral toxicity studies, respectively, did not exhibit any abnormal behaviors or mortality. There were only minor changes in body weight, organ weight, hematological and blood chemistry values. However, gross pathological and histopathological studies revealed no abnormalities in the internal organs. Therefore, it can be concluded that Maha Pigut Triphala in the ratio of 2:1:3 causes neither acute nor chronic oral toxicity in rats. Additionally, the present study provides safety information on Maha Pigut Triphala in animals that could help guide future studies in humans.

## 1. Introduction

The fruits of many plants such as *T. bellirica* (Family Combretaceae), *T. chebula* (Family Combretaceae), and *E. officinalis* or *P. emblica* (Family Euphorbiaceae) can be used as food, dietary supplements, or herbal medicines. The dried fruits of these plants are often prepared as extract powder for dietary supplements or herbal formulas to prevent and treat diseases [1,2,3]. Traditional medicine is an alternative option that people in developing countries use to treat or relieve different diseases and improve their health [4]. One of the best-known is Maha Pigut Triphala, a blend of the dried fruits of *T. bellirica*, *T. chebula*, and *E. officinalis*, respectively. In traditional Thai medicine, Maha Pigut Triphala is prepared in one of three different formulas. The first formula is called Vata Samutthan, which has a ratio of 1:3:2. This formula is prescribed for use during the rainy season to relieve dizziness, muscle aches, and stomach cramps. The second formula, which is known as Pitta Samutthan, has a ratio of 3:2:1. This formula is used to treat fever and headache during the summer season. The third formula, namely Semha Samutthan, has a ratio of 2:1:3. It is prescribed to relieve colds, runny nose, phlegm, and diarrhea in the winter season [5]. The dried fruits of *T. bellirica*, *T. chebula*, and *E. officinalis* or *P. emblica* were prepared in a ratio of 1:1:1 respectively, which is known as the Triphala recipe in traditional Ayurvedic medicine [6,7]. The Triphala recipe is commonly used to balance digestion, eliminate waste from the body, and act as a laxative [8,9]. Several studies, both in vitro and in vivo experiments, have reported on the pharmacological effects of Maha Pigut Triphala and the Triphala recipe, such as antioxidant [10], antitumor [11], antidiabetic [12], anti-hyperlipidemic [13], anti-inflammatory [14] and immunomodulatory activities [15].

In addition to studying the effectiveness of herbal treatment, safety from using herbs is another important aspect to consider. Therefore, herbal toxicity testing is necessary to be carried out. The study and development of any manufactured drugs or medicinal plants normally requires testing of toxicity in laboratory animals. The primary purpose is to assess the safety profile of these substances, which will be critical information for further clinical testing in humans [16]. In an acute oral toxicity test, experimental animals are normally evaluated after administration of a single large oral dose of the test substance or multiple smaller doses over a 24 h period [17]. The fact that a test substance does not show immediate toxicity does not mean that it cannot cause long-term toxicity because the substance may need to accumulate over time for it to exhibit its toxicity. Accordingly, it is necessary to perform a chronic oral toxicity test by giving test substances to experimental animals for up to 9–12 months to determine the effect on various organs as well as to evaluate hematological and blood chemistry parameters that the substance may influence [17,18].

Because both Maha Pigut Triphala and the Triphala recipe have been used in traditional medicine for a long time, it is considered to be highly safe. An acute toxicity study revealed no mortality occurred in male Swiss albino mice administered an aqueous extract of Triphala recipe (1:1:1) at a concentration of 240 mg/kg body weight [19]. A recent study exhibited that a single oral dose of the Triphala recipe (1:1:1) at a concentration of 5000 mg/kg body weight did not cause acute toxicity in Sprague-Dawley rats. Moreover, rats that received the Triphala recipe (1:1:1) orally at concentrations of up to 2400 mg/kg body weight for 270 days in a chronic toxicity test showed no abnormalities or mortality [20].

In the present study, Maha Pigut Triphala, which consisted of *T. bellirica*, *T. chebula*, and *E. officinalis,* was prepared in the ratio of 2:1:3, respectively. More importantly, the results of toxicity testing of this formula have not been reported before. Thus, the present study aimed to evaluate acute and chronic oral toxicity testing of Maha Pigut Triphala (2:1:3) in Sprague-Dawley rats.

## 2. Materials and Methods

### 2.1. Plant Material and Extract Preparation

The dried fruits of *T. bellirica* and *T. chebula* were purchased from Vejpong Osot Pharmacy, Bangkok, Thailand. The dried fruits of *E. officinalis* were collected from their natural habitats, located in Nan Province, Thailand. Quality control of the plant materials and Maha Pigut Triphala was carried out by evaluating the percentage weight loss on drying, extractive values, total ash, and acid insoluble ash according to the method in Thai Herbal Pharmacopoeia [21].

All three plants were extracted using the same method. Firstly, the dried fruits were boiled in water for 1 h, repeating this process 3 times. After that, the residue was filtered and then the filtrate was evaporated by a spray dryer to produce the extract in powder form. Finally, each type of plant extract powder consisting of *T. bellirica*, *T. chebula*, and *E. officinalis* was combined in a ratio of 2:1:3, respectively, to obtain Maha Pigut Triphala and it was stored at −20 °C and dissolved in distilled water before giving it to the rats.

### 2.2. Thin Layer Chromatography (TLC)

The quality control of raw materials and plant extracts in Maha Pigut Triphala was examined by thin-layer chromatography using modified methods according to Cordeiro et al. [22]. Ellagic acid, gallic acid, and protocatechuic acid were used as reference standards. All samples were applied on a sheet of aluminum coated with silica gel 60 GF254 (Merck, Darmstadt, Germany), and the chromatogram was developed using a mixture of ethyl acetate:formic acid:water (8:1:1) as a mobile phase. Chromatoplates were dried and visualized under UV light at 254 and 366 nm or sprayed with anisaldehyde-sulfuric acid and phosphomolybdic acid (Figure 1).

### 2.3. Experimental Animals

Male and female Sprague-Dawley rats (weighing 180–200 g) were obtained from the National Laboratory Animal Center, Mahidol University, Nakhon Pathom, Thailand. They were housed in ventilated cages at a room temperature of 25 ± 1 °C, 60% relative humidity, and under a 12 h light/12 h dark cycle. A standard pelleted diet and drinking water were provided ad libitum. The rats were acclimated for at least 1 week before the experiments. All animal procedures were approved by the Animal Ethical Committee of the Faculty of Medicine, Thammasat University, Thailand (AE004/2552).

### 2.4. Acute Oral Toxicity of Maha Pigut Triphala

The acute oral toxicity test was performed in accordance with Co-operation and Development (OECD) Test Guideline 420 and World Health Organization (WHO) Guidelines [23,24]. Ten female rats were randomly assigned to 2 groups of 5 rats/group. The rats in the control group were orally gavaged with distilled water (1 mL/kg body weight), whereas the test group received Maha Pigut Triphala at the dose of 5000 mg/kg body weight. Hippocratic screening was a procedure used to evaluate the acute toxicity of Maha Pigut Triphala at 5, 10, 15, 30, 60, 120, and 240 min after administration and every hour for 24 h. Hippocratic screening was performed by scoring distinctive parameters, consisting of a decrease in motor activity and respiration rate and a loss of righting reflex and screen grip [25]. The animals were individually observed for signs of significant changes such as sedation, vomiting, muscle spasms, fatigue, and watery diarrhea within 24 h after administration and once daily for 14 days. Furthermore, the body weight of rats and the presence of mortality were recorded throughout the experimental period.

At the end of the study, the rats were sacrificed by intraperitoneal injection of pentobarbital sodium (40 mg/kg body weight); after that, the vital signs, reflexes, and pulse were verified to confirm death [26]. Gross pathological changes of various internal organs, including the heart, lungs, liver, pancreas, kidneys, spleen, adrenal glands, stomach, intestines, ovaries, uterus, eyes, muscles, bones, and brain, were observed by the naked eye, and any internal organs with abnormalities were further examined in the histopathological study.

### 2.5. Chronic Oral Toxicity of Maha Pigut Triphala

The chronic oral toxicity test was conducted based on OECD Test Guideline 452 and the WHO Guidelines [18,24]. Ninety rats were divided into 5 groups (10 rats/sex/group in groups 1–4 and 5 rats/sex/group in group 5). The control group (group 1) was given distilled water with a dose of 1 mL/kg body weight. The three groups (groups 2–4) were test groups and were administered 600, 1200, and 2400 mg/kg body weight of Maha Pigut Triphala, respectively. The satellite group (group 5) received Maha Pigut Triphala at the dose of 2400 mg/kg body weight. The test substances were administered orally once daily for 270 consecutive days. During the experiment period, the rats were weighed, and any behavior changes or disorder symptoms were observed and recorded by the Hippocratic screening protocol [25]. The rats that died during the experiment were necropsied for gross pathological examination, and abnormalities of internal organs were additionally investigated by histopathological study.

On day 270, the rats in groups 1–4 were sacrificed by intraperitoneal injection of pentobarbital sodium (40 mg/kg body weight) [26], whereas the rats in group 5 (satellite group) that received Maha Pigut Triphala for 270 days, were raised for another 28 days without receiving the test substance before being euthanized. At the end of the experiment, the animal’s death was confirmed by checking the vital signs, reflexes, and pulse [26]. Blood samples were obtained via cardiac puncture and collected into tubes containing ethylenediaminetetraacetic acid (EDTA) and serum clot activator tubes for examination of hematology and blood chemistry, respectively [27]. The blood samples in the EDTA-containing tubes were analyzed using a Mindray BC-5300 Vet automated hematology analyzer (Shenzhen, China). The blood samples in the serum clot activator tubes were centrifuged at 3500 rpm for 10 min to obtain serum for analysis using an automated BX-3010 analyzer (Sysmex, Kobe, Japan). A gross necropsy was performed to visually examine internal organ abnormalities (brain, lungs, heart, liver, pancreas, spleen, adrenal glands, kidneys, ovaries, uterus, testes, and epididymis), and these organs were weighted and then preserved in 10% formaldehyde for histopathological examination with Hematoxylin & Eosin (H&E) stain.

### 2.6. Statistical Analysis

The data are expressed in terms of mean ± standard error of the mean (S.E.M) and analyzed using SPSS statistics software version 25 (SPSS Inc., Chicago, IL, USA). A *t*-test or a Mann–Whitney U test was performed to test the difference in parametric or nonparametric data, respectively, of the acute oral toxicity study. A Shapiro–Wilk test was used to investigate the data normality of the chronic oral toxicity study. Then, for normally distributed data, a comparison of the results between groups was analyzed using a one-way analysis of variance (ANOVA) followed by Tukey’s multiple comparison tests. On the contrary, for non-normally distributed data, a Kruskal–Wallis nonparametric ANOVA test followed by a Dunn’s test was used to identify any difference among the groups. A *p*-value less than 0.05 was considered statistically significant.

## 3. Results

### 3.1. Effect of Acute Oral Toxicity Study of Maha Pigut Triphala

Maha Pigut Triphala was orally administered at a dose of 5000 mg/kg to female rats, then assessed for potential toxicity of Maha Pigut Triphala within the first 24 h and closely observed every day for 14 days by Hippocratic screening (Appendix A). Observation of the rats’ symptoms and behaviors during the study period of treatment revealed that the female rats had no abnormal symptoms, behavioral changes, or mortality that could indicate Maha Pigut Triphala toxicity. The effect of Maha Pigut Triphala on the body weight of female rats is shown in Figure 2.

After 14 days, the rats were sacrificed, and the carcasses were dissected to examine the abnormalities of various tissues and organs, including the heart, lungs, liver, pancreas, kidneys, spleen, adrenal glands, stomach, intestines, ovaries, uterus, eyes, muscles, bones, and brains. The gross pathological examination of internal organs with the naked eye revealed no organ lesions or size, shape, or color abnormalities in female rats administered with Maha Pigut Triphala at 5000 mg/kg compared to the control group.

### 3.2. Effect of Chronic Oral Toxicity Study of Maha Pigut Triphala

The chronic oral toxicity was tested by giving oral Maha Pigut Triphala doses of 600, 1200, and 2400 mg/kg to female and male rats once daily for 270 days. Observing the symptoms of rats after receiving Maha Pigut Triphala did not reveal behavioral changes, abnormalities, or mortality in rats of both sexes (Appendix A).

The effect of Maha Pigut Triphala on the body weight of female and male rats in the chronic oral toxicity study is shown in Figure 3 and Figure 4, respectively. It was found that the female rats receiving Maha Pigut Triphala at a dose of 600 mg/kg had a significant decrease in body weight on day 30. The female rats receiving Maha Pigut Triphala at a dose of 1200 mg/kg had a significant increase in body weight on days 90 and 180 compared to the control group. Moreover, a noticeable increase in body weight was found in female rats in the 2400 mg/kg Maha Pigut Triphala-treated group on day 180 compared to the control group. In comparison, the male rats in both the 2400 mg/kg Maha Pigut Triphala-treated group and the satellite group had a significant increase in body weight on day 180 compared to the control group.

The organ weights of female and male rats administered with different concentrations of Maha Pigut Triphala for 270 days are presented in Table 1 and Table 2, respectively. The female rats administered with Maha Pigut Triphala at a dose of 1200 mg/kg had a markedly increased spleen weight. On the other hand, a significant decrease in spleen weight was observed in the satellite group of female rats when compared to the control group. The group of female rats receiving Maha Pigut Triphala at a dose of 1200 and 2400 mg/kg, as well as the female rats in the satellite groups, exhibited a significant decrease in kidney weight as compared to the control group. In the male rats, brain weight notably decreased in the group receiving Maha Pigut Triphala at a dose of 600 mg/kg. The 2400 mg/kg Maha Pigut Triphala-treated male rats had remarkably increased lung and heart weights but decreased testes weight. Additionally, significantly increased heart weight and decreased pancreas weight were shown in the male rats in the satellite group when compared to the control group. All of the internal organs were subjected to rigorous and detailed gross pathological and histopathological analysis to determine if there were any abnormalities, especially the internal organs that had significantly increased or decreased in weight.

The hematology values of female rats in the chronic toxicity study are displayed in Table 3. The female rats that received Maha Pigut Triphala doses of 600 and 1200 mg/kg revealed a marked increase in the mean corpuscular volume (MCV). As for the male rats in the satellite group, the number of platelets (PLT) significantly decreased as compared to the control group, and the results are exhibited in Table 4.

The differential WBC counts of the female rats are shown in Table 5. It was found that the female rats given Maha Pigut Triphala doses of 600 and 1200 mg/kg, as well as the female rats in the satellite group, had a notably decreased number of lymphocytes (LYMPs). Table 6 demonstrates the differential WBC counts of the male rats. The results showed a significant increase in the number of neutrophils (NEUs) in male rats given Maha Pigut Triphala at a dose of 600 mg/kg. WBC and LYMP counts were markedly increased in male rats in the 2400 mg/kg Maha Pigut Triphala-treated group. Moreover, a significant increase in WBC, NEU, and LYMP counts was observed in male rats in the satellite group. The increase or decrease in the differential WBC counts was statistically significantly different compared to the control group.

The blood chemistry values of female and male rats are shown in Table 7 and Table 8, respectively. The results revealed that the female rats in the 2400 mg/kg Maha Pigut Triphala-treated group had a notable decrease in the levels of serum glutamic oxaloacetate transaminase (SGOT). The female rats in the satellite group exhibited a significant increase in total bilirubin levels as compared to the control group. Furthermore, the blood glucose levels of the male rats that received 1200 mg/kg of Maha Pigut Triphala were remarkably decreased compared to the control group. It was found that the male rats in the 2400 mg/kg Maha Pigut Triphala-treated group had significantly lower albumin levels. In addition, a significant decrease in total protein levels and a marked increase in total bilirubin levels were observed in the satellite group of male rats compared to the control group.

The study of the gross pathology and histopathology of internal organs, including the brain, lungs, heart, liver, pancreas, spleen, adrenal glands, kidneys, ovaries, uterus, testes, and epididymis did not reveal any abnormalities in all internal organs of both female and male rats administered with Maha Pigut Triphala at various concentrations as compared to the control group (Figure 5 and Figure 6).

## 4. Discussion

Medicinal plants are considered a rich source of natural bioactive compounds with medicinal properties, and they have been used as both dietary supplements and herbal medicines in traditional medicine in Asia for a long time [28]. Maha Pigut Triphala, one of the well-known formulas in traditional Thai medicine, consists of various proportions of dried fruits of *T. bellirica*, *T. chebula*, and *E. officinalis*, depending on the therapeutic purpose of treating different seasonal illnesses as prescribed by traditional Thai medicine doctors [5]. The three plants that are the components of Maha Pigut Triphala have been previously studied for their various pharmacological activities and have been found to have antioxidant, antidiabetic, anti-inflammatory, antimicrobial, hepatoprotective, and immunomodulatory properties [29,30,31].

In addition to studying the pharmacological effects of medicinal plants, evaluating their toxicity is equally important. Such evaluations provide essential information on the safety of medicinal plants when tested in laboratory animals, which is a necessary step before human trials for potential future use [16]. The toxicity of the plants composing Maha Pigut Triphala has been examined previously. The results showed that the aqueous extracts from fruits of three plants, namely *T. bellirica*, *T. chebula*, and *E. officinalis*, when administered to rats either as a single oral dose of 5000 mg/kg or at a concentration of up to 1200 mg/kg/day over 270 consecutive days in acute and chronic toxicity studies, respectively, caused no abnormalities or mortality, thereby demonstrating the safety of these extracts [32,33,34]. Moreover, neither acute nor chronic toxicity was found in rats treated with a Triphala recipe (1:1:1) in recent research [20]. These findings prompted the current study, which aimed to evaluate the acute and chronic oral toxicity of Maha Pigut Triphala prepared in a 2:1:3 ratio.

The acute oral toxicity test in the present study proceeded according to OECD Test Guideline 420 and the WHO Guidelines [23,24]. A high dose of Maha Pigut Triphala, 5000 mg/kg body weight, was given as a single oral dose. Then, the animal’s condition was observed for the first 24 h and throughout the 14 days of the experiment. Important observations include vomiting, convulsions, muscle tremors, muscle weakness, rapid heart rate, difficulty breathing, blood in either urine or feces, abnormal skin color, and hair loss [35]. This study found no abnormal symptoms, indicating that Maha Pigut Triphala caused acute toxicity in female rats. All rats remained healthy and did not show any signs of illness. Furthermore, necropsy and gross pathology revealed no abnormalities of internal organs. Thereby, it can be inferred that Maha Pigut Triphala orally administered to female rats at a single dose of 5000 mg/kg body weight does not cause acute toxicity.

A chronic oral toxicity study was conducted to evaluate the effects of continuous administration of a test substance in laboratory animals over a period of 9–12 months following OECD Test Guideline 452 and the WHO Guidelines [18,24]. In the present study, Maha Pigut Triphala was administered at doses of 600, 1200, and 2400 mg/kg body weight once daily for a period of 270 days. Throughout the experiment, animal symptoms and appearances were observed and health examinations were performed to assess the potential toxicity of the test substances. The results showed that no abnormal symptoms, behavioral changes, or illnesses occurred in the rats that received Maha Pigut Triphala, indicating that Maha Pigut Triphala caused no toxicity in both male and female rats, including the satellite groups of male and female rats, which followed up on the effects of Maha Pigut Triphala after 270 days of administration and continued observation for 28 days beyond the administration of Maha Pigut Triphala.

Changes in body and organ weight are critical parameters in evaluating the toxicity of a test substance. Deviations from the control group or reference values may indicate potential toxic effects on the body’s systems [36,37]. Thus, on the 270th day of the experiment, the body weight and organ weights of the rats were compared with the control group. It was found that the body weights of the rats of both sexes that received Maha Pigut Triphala were not different from those of the control group, whereas significant differences in some organs, including the heart, spleen, and kidney, were exhibited in both male and female rats. However, these statistical differences in organ weights are still within the normal range of reference values [38]. Therefore, these differences might be the results of individual variations in animals, not the effect of the test substances. According to the results of body weight and organ weight, Maha Pigut Triphala did not cause chronic toxicity in rats.

Hematology is an important study for the diagnosis of abnormalities in various systems in laboratory animals when exposed to test substances for a long period of time, and it can also be used to evaluate symptoms of disease that may occur. The components of blood include red blood cells (RBCs), WBCs, and PLTs. These parameters can be used to evaluate the health status of laboratory animals. As assessed by the RBC count, the values below the normal range might indicate anemia, massive blood loss, or abnormalities of erythropoiesis and oxygen transport. On the other hand, the values above the normal range point to excessive blood concentration, which can result from oxygen starvation and dehydration. Additionally, WBC counts can be used to indicate infection, inflammation, immune system disorders, and secretory system disorders, while PLT levels might be used to evaluate the immune system and blood clotting system [39,40,41]. The present study has revealed that some hematological parameters of male and female rats treated with Maha Pigut Triphala, including the MCV, PLT, WBC, NEU, and LYMP count, were statistically different when compared to the control group. This differs from a previous chronic toxicity study that found that the number of RBCs, hemoglobin (HB), hematocrit (HCT), MCV, and NEU decreased significantly in male rats administered with high doses of the Triphala recipe [20]. Nonetheless, when compared to the reference ranges established by the National Laboratory Animal Center [42,43], the hematological values in this study remained within normal ranges. Therefore, it can be inferred that Maha Pigut Triphala at doses of 600, 1200, and 2400 mg/kg body weight does not cause any abnormalities in the circulatory system or blood cell formation, as well as no effect on the immune system of the animals.

The functionality of the pancreas, kidneys, and liver can be assessed by blood chemistry tests. The blood chemistry parameters can be used to indicate the physiological state of vital organs or to evaluate pathological conditions occurring in laboratory animals [44]. The measurement of blood glucose levels is a preliminary indicator of pancreatic function because if there is an increase in blood glucose levels after exposure to the test substance, it might be a result of the test substance destroying the pancreatic cells [45]. Total protein, albumin, bilirubin, alkaline phosphatase (ALP), SGOT, and serum glutamate pyruvate transaminase (SGPT) are common indicators of liver function and pathology. Total protein and albumin concentration are used to evaluate the functioning of liver cells because these two substances are the most produced in the liver. Therefore, whenever liver abnormalities or damage occur, there will be a change in these two values [46].

Bilirubin, a waste product from the breakdown of hemoglobin in expired or damaged red blood cells, exists in two forms: unconjugated (indirect) bilirubin, which is transported to the liver, and conjugated (direct) bilirubin, processed in the liver. An increase in indirect bilirubin levels might point to hemolysis or the destruction of red blood cell precursors. In addition, an increase in the levels of direct bilirubin might indicate a blockage within the liver, such as cirrhosis or hepatitis. It might also be caused by a bile duct obstruction from the pancreas due to certain medications, or a blockage outside the liver, such as a common bile duct stone [47].

In the measurements of liver enzymes, higher ALP levels can indicate an obstruction of the liver, whereas elevated levels of SGOT and SGPT are signs of liver cell damage [48]. When comparing the blood chemistry parameters of all groups of male and female rats that received various concentrations of Maha Pigut Triphala with the control group, the results showed that some blood chemistry parameters were statistically different, including glucose, total protein, albumin, total bilirubin, and SGOT. This is consistent with another study using a Triphala recipe [20]. Nevertheless, the slight differences in elevation or decrease in these blood chemistry parameters did not indicate any impairment of pancreas, kidney, or liver function, and all these values were still within the normal range of the reference values [49]. Furthermore, in the present research, neither physical nor behavioral abnormalities were found that could indicate that there was an occurrence of physiological disorders in laboratory animals, and pathological abnormalities of the vital organs, including the pancreas, kidney, and liver.

A study on the histopathological abnormalities of tissues and organs in all groups of rats treated with Maha Pigut Triphala compared to the control group is the final step, which is a key element to indicate whether the test substance can cause chronic toxicity [50]. From gross pathological and histopathological examinations of the internal organs, including the brain, lungs, heart, liver, pancreas, spleen, adrenal glands, kidneys, ovaries, uterus, testes, and epididymis, no gross pathological lesions or abnormal size, shape, and color of organs were found in Maha Pigut Triphala-treated groups compared to the control group. Similar results were seen in a previous study that used a Triphala recipe [20]. Altogether, the results showed that Maha Pigut Triphala was safe and did not cause chronic toxicity when administered to both female and male rats for a consecutive period of 270 days. In addition, no toxic effects were observed for a period of up to 28 days post-administration of Maha Pigut Triphala.

## 5. Conclusions

In summary, the results of acute and chronic oral toxicity studies suggest that Maha Pigut Triphala in the ratio of 2:1:3 does not cause toxicity. Neither a single oral dose of Maha Pigut Triphala at a high dose (5000 mg/kg body weight) nor daily feeding of Maha Pigut Triphala for 270 days produced any signs and symptoms of toxicity in rats.

## Figures and Tables

**Figure 1 biology-13-01005-f001:**
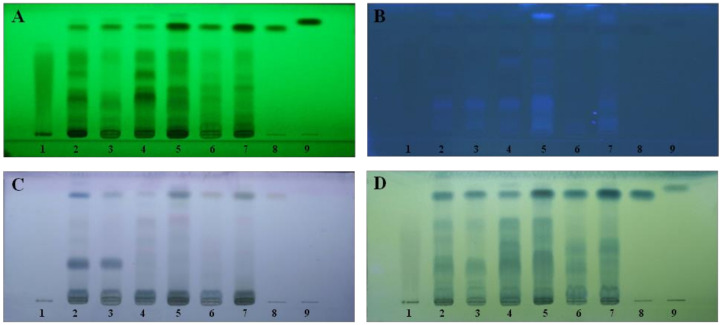
TLC chromatograms of compounds after being visualized under 254 nm (**A**) and 366 nm (**B**) UV light or sprayed with anisaldehyde-sulfuric acid (**C**) and phosphomolybdic acid (**D**); 1 = ellagic acid, 2 = aqueous extract of *T. chebula*, 3 = raw material of *T. chebula*, 4 = aqueous extract of *T. bellirica*, 5 = raw material of *T. bellirica*, 6 = aqueous extract of *E. officinalis*, 7 = raw material of *E. officinalis*, 8 = gallic acid, 9 = protocatechuic acid.

**Figure 2 biology-13-01005-f002:**
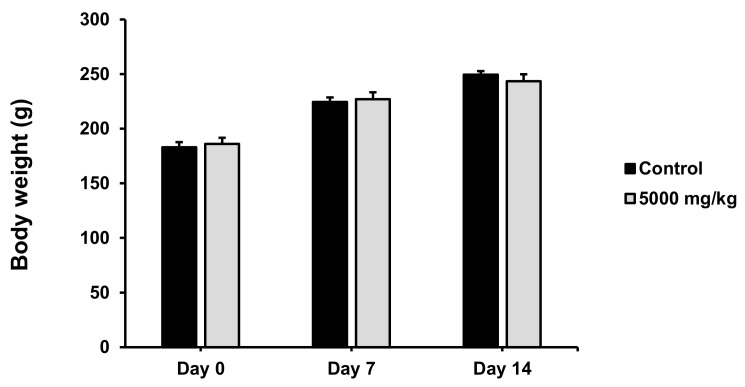
Body weight of female rats in the acute oral toxicity study of Maha Pigut Triphala. Results are expressed as mean ± S.E.M., *n* = 5.

**Figure 3 biology-13-01005-f003:**
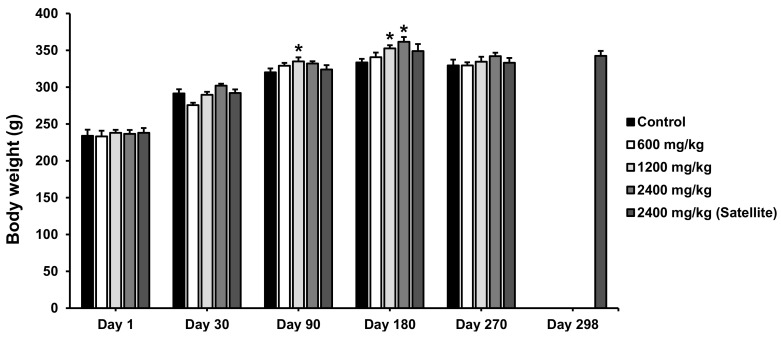
Body weight of female rats in the chronic oral toxicity study of Maha Pigut Triphala. Results are expressed as mean ± S.E.M., *n* = 10, *n* = 5 (satellite group). * *p* < 0.05 compared with the control group.

**Figure 4 biology-13-01005-f004:**
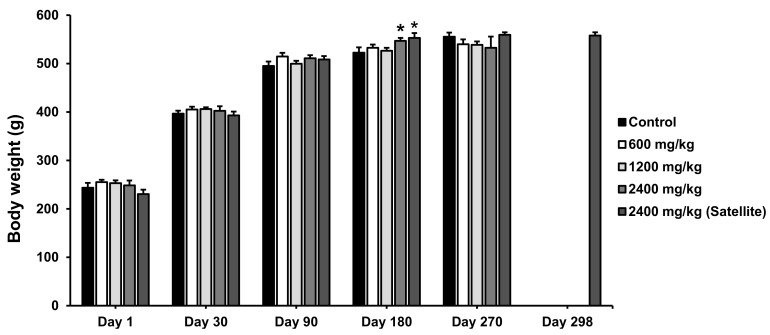
Body weight of male rats in the chronic oral toxicity study of Maha Pigut Triphala. Results are expressed as mean ± S.E.M., *n* = 10, *n* = 5 (satellite group). * *p* < 0.05 compared with the control group.

**Figure 5 biology-13-01005-f005:**
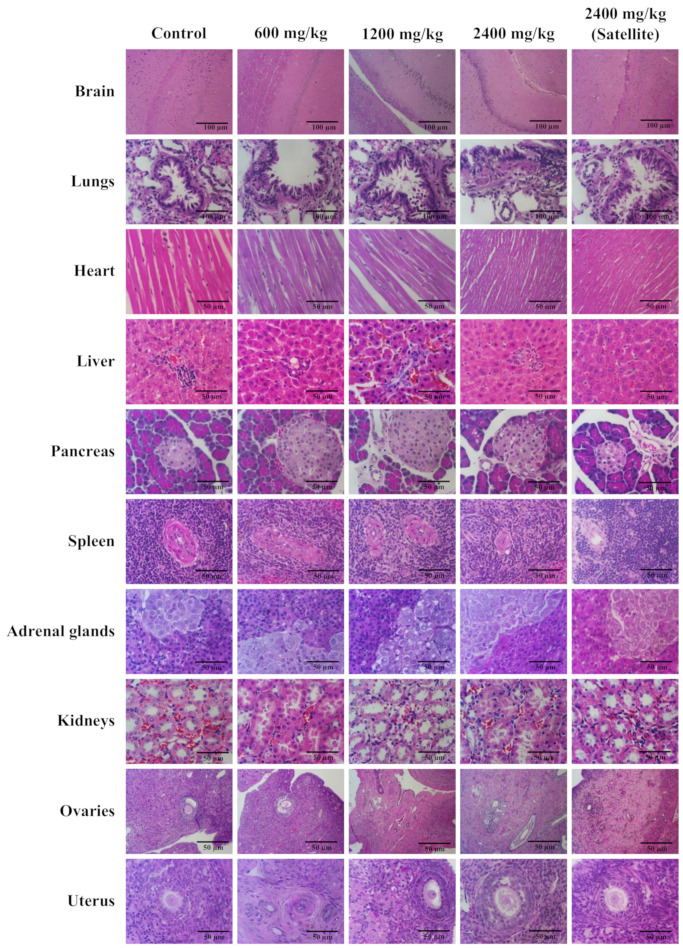
The histological structure in H&E-stained images of different types of tissues from female rats in the study of chronic oral toxicity of Maha Pigut Triphala. (Scale bars, 100 μM; 10× magnification, 50 μM; 40× magnification).

**Figure 6 biology-13-01005-f006:**
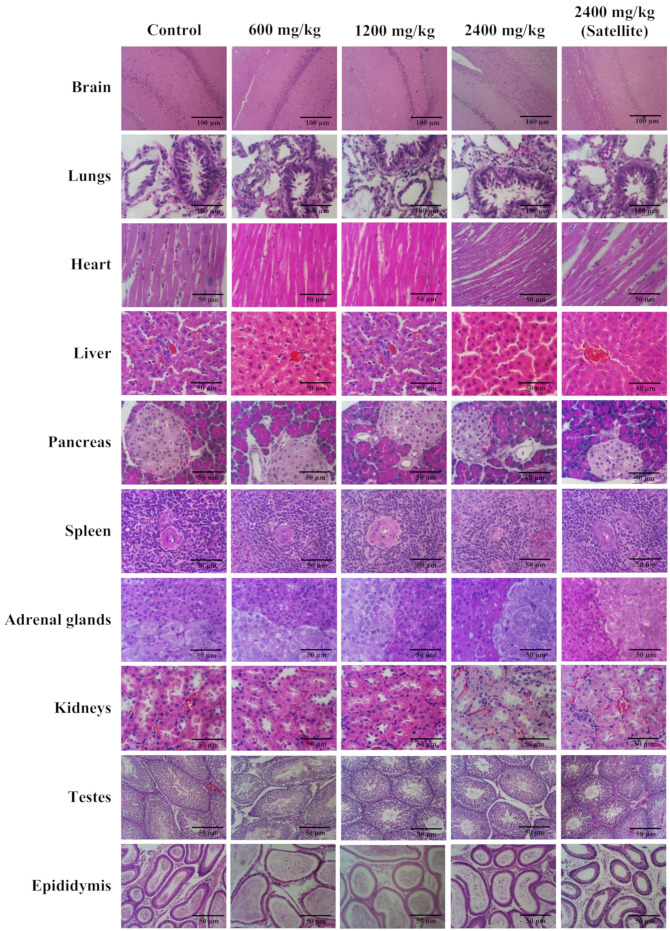
The histological structure in H&E-stained images of different types of tissues from male rats in the study of chronic oral toxicity of Maha Pigut Triphala. (Scale bars, 100 μM; 10× magnification, 50 μM; 40× magnification).

**Table 1 biology-13-01005-t001:** Organ weight of female rats in the chronic oral toxicity study of Maha Pigut Triphala.

Organs	Control	Maha Pigut Triphala (mg/kg)			
		600	1200	2400	2400 (Satellite)
Brain	1.94 ± 0.02	1.93 ± 0.03	1.98 ± 0.03	1.91 ± 0.03	1.90 ± 0.02
Lungs	2.18 ± 0.14	2.16 ± 0.15	2.43 ± 0.24	2.08 ± 0.15	2.00 ± 0.22
Heart	1.24 ± 0.03	1.31 ± 0.03	1.30 ± 0.03	1.29 ± 0.03	1.35 ± 0.06
Liver	9.12 ± 0.88	10.12 ± 0.21	9.84 ± 0.46	10.06 ± 0.23	10.17 ± 0.24
Pancreas	1.24 ± 0.14	1.04 ± 0.05	1.00 ± 0.08	1.03 ± 0.05	0.86 ± 0.04
Spleen	0.81 ± 0.04	0.81 ± 0.04	0.84 ± 0.04 *	0.78 ± 0.05	0.76 ± 0.02 *
Adrenal glands	0.04 ± 0.00	0.06 ± 0.02	0.04 ± 0.00	0.04 ± 0.00	0.03 ± 0.00
Kidneys	1.32 ± 0.02	1.29 ± 0.02	1.25 ± 0.03 *	1.25 ± 0.02 *	1.24 ± 0.02 *
Ovaries	0.06 ± 0.00	0.06 ± 0.00	0.06 ± 0.01	0.07 ± 0.00	0.06 ± 0.01
Uterus	1.36 ± 0.31	1.34 ± 0.17	1.51 ± 0.34	2.71 ± 0.53	2.31 ± 0.81

Results are expressed as mean ± S.E.M., *n* = 10, *n* = 5 (satellite group). * *p* < 0.05 compared with the control group.

**Table 2 biology-13-01005-t002:** Organ weight of male rats in the chronic oral toxicity study of Maha Pigut Triphala.

Organs	Control	Maha Pigut Triphala (mg/kg)			
		600	1200	2400	2400 (Satellite)
Brain	2.12 ± 0.02	2.03 ± 0.03 *	2.05 ± 0.04	2.04 ± 0.03	2.12 ± 0.02
Lungs	2.74 ± 0.31	2.79 ± 0.19	2.73 ± 0.22	3.52 ± 0.36 *	2.70 ± 0.13
Heart	1.78 ± 0.03	1.70 ± 0.04	1.82 ± 0.06	1.96 ± 0.08 *	2.09 ± 0.04 *
Liver	17.79 ± 0.55	17.02 ± 0.92	16.77 ± 0.55	17.38 ± 1.11	18.57 ± 0.76
Pancreas	1.29 ± 0.04	1.22 ± 0.09	1.29 ± 0.05	1.15 ± 0.10	0.93 ± 0.10 *
Spleen	1.04 ± 0.05	1.02 ± 0.04	1.00 ± 0.04	1.02 ± 0.06	1.05 ± 0.03
Adrenal glands	0.04 ± 0.00	0.03 ± 0.00	0.03 ± 0.00	0.03 ± 0.00	0.04 ± 0.00
Kidneys	1.71 ± 8.70	1.88 ± 0.05	1.92 ± 0.06	1.87 ± 0.06	2.05 ± 0.04
Testes	2.17 ± 0.04	2.14 ± 0.02	2.12 ± 0.03	2.07 ± 0.05 *	2.08 ± 0.03
Epididymis	0.84 ± 0.01	0.85 ± 0.01	0.85 ± 0.02	0.83 ± 0.03	0.84 ± 0.01

Results are expressed as mean ± S.E.M., *n* = 10, *n* = 5 (satellite group). * *p* < 0.05 compared with the control group.

**Table 3 biology-13-01005-t003:** Hematological parameters of female rats in the chronic oral toxicity study of Maha Pigut Triphala.

Parameters	Control	Maha Pigut Triphala (mg/kg)			
		600	1200	2400	2400 (Satellite)
RBC (×10^6^/µL)	8.55 ± 0.22	8.64 ± 0.11	8.17 ± 0.63	8.49 ± 0.16	8.93 ± 0.11
HB (g/dL)	18.47 ± 3.18	15.84 ± 0.20	14.62 ± 1.04	15.42 ± 0.19	15.66 ± 0.12
HCT (%)	46.16 ± 3.68	51.10 ± 0.72	48.60 ± 3.72	49.90 ± 0.77	51.00 ± 0.54
MCV (fl)	57.80 ± 0.57	59.10 ± 0.31 *	59.30 ± 0.42 *	58.90 ± 0.50	57.30 ± 0.37
MCH (pg)	17.86 ± 0.21	18.35 ± 0.18	18.03 ± 0.30	18.18 ± 0.18	17.62 ± 0.11
MCHC (g/dL)	30.82 ± 0.16	30.97 ± 0.26	30.43 ± 0.54	30.89 ± 0.22	30.83 ± 0.15
PLT (×10^6^/µL)	0.83 ± 0.03	0.79 ± 0.02	0.87 ± 0.02	0.81 ± 0.03	0.78 ± 0.02

Results are expressed as mean ± S.E.M., *n* = 10, *n* = 5 (satellite group). * *p* < 0.05 compared with the control group. RBC, red blood cell; HB, hemoglobin; HCT, hematocrit; MCV, mean corpuscular volume; MCH, mean corpuscular hemoglobin; MCHC, mean corpuscular hemoglobin concentration; PLT, platelet.

**Table 4 biology-13-01005-t004:** Hematological parameters of male rats in the chronic oral toxicity study of Maha Pigut Triphala.

Parameters	Control	Maha Pigut Triphala (mg/kg)			
		600	1200	2400	2400 (Satellite)
RBC (×10^6^/µL)	9.53 ± 0.22	9.58 ± 0.36	9.44 ± 0.31	9.77 ± 0.30	10.15 ± 0.24
HB (g/dL)	16.32 ± 0.36	16.68 ± 0.34	16.53 ± 0.23	16.61 ± 0.34	16.75 ± 0.34
HCT (%)	53.70 ± 1.05	54.90 ± 1.55	53.90 ± 1.73	55.60 ± 1.62	55.80 ± 1.28
MCV (fl)	56.20 ± 0.70	57.60 ± 0.70	57.20 ± 0.71	57.00 ± 0.80	54.60 ± 0.40
MCH (pg)	17.13 ± 0.17	17.52 ± 0.41	17.67 ± 0.67	17.02 ± 0.23	16.45 ± 0.14
MCHC (g/dL)	30.46 ± 0.33	30.39 ± 0.39	30.89 ± 0.90	29.91 ± 0.33	30.07 ± 0.13
PLT (×10^6^/µL)	0.94 ± 0.03	0.97 ± 0.07	0.86 ± 0.05	0.91 ± 0.04	0.77 ± 0.03 *

Results are expressed as mean ± S.E.M., *n* = 10, *n* = 5 (satellite group). * *p* < 0.05 compared with the control group. RBC, red blood cell; HB, hemoglobin; HCT, hematocrit; MCV, mean corpuscular volume; MCH, mean corpuscular hemoglobin; MCHC, mean corpuscular hemoglobin concentration; PLT, platelet.

**Table 5 biology-13-01005-t005:** Differential WBC counts in the peripheral blood of female rats in the chronic oral toxicity study of Maha Pigut Triphala.

Parameters	Control	Maha Pigut Triphala (mg/kg)			
		600	1200	2400	2400 (Satellite)
WBC (×10^3^/µL)	5.28 ± 1.31	3.17 ± 0.14	4.44 ± 1.17	4.44 ± 0.22	3.76 ± 0.35
NEU (×10^3^/µL)	1.46 ± 0.78	0.78 ± 0.15	1.73 ± 0.94	1.09 ± 0.10	1.07 ± 0.20
LYMP (×10^3^/µL)	3.66 ± 0.51	2.28 ± 0.19 *	2.56 ± 0.29 *	3.25 ± 0.25	2.61 ± 0.23 *
MONO (×10^3^/µL)	0.14 ± 0.04	0.32 ± 0.22	0.12 ± 0.04	0.10 ± 0.03	0.08 ± 0.01
EO (×10^3^/µL)	0.03 ± 0.02	0.02 ± 0.00	0.03 ± 0.02	0.01 ± 0.01	0.02 ± 0.01
BASO (×10^3^/µL)	0.00 ± 0.00	0.00 ± 0.00	0.00 ± 0.00	0.00 ± 0.00	0.00 ± 0.00

Results are expressed as mean ± S.E.M., *n* = 10, *n* = 5 (satellite group). * *p* < 0.05 compared with the control group. WBC, white blood cell; NEU, neutrophil; LYMP, lymphocyte; MONO, monocyte; EO, eosinophil; BASO, basophil.

**Table 6 biology-13-01005-t006:** Differential WBC counts in the peripheral blood of male rats in the chronic oral toxicity study of Maha Pigut Triphala.

Parameters	Control	Maha Pigut Triphala (mg/kg)			
		600	1200	2400	2400 (Satellite)
WBC (×10^3^/µL)	5.05 ± 0.53	5.54 ± 0.44	5.29 ± 0.34	7.09 ± 0.27 *	7.86 ± 0.56 *
NEU (×10^3^/µL)	0.81 ± 0.05	1.62 ± 0.34 *	1.25 ± 0.23	1.45 ± 0.22	1.70 ± 0.27 *
LYMP (×10^3^/µL)	4.12 ± 0.51	3.83 ± 0.29	3.88 ± 0.34	5.42 ± 0.27 *	6.02 ± 0.47 *
MONO (×10^3^/µL)	0.12 ± 0.04	0.09 ± 0.02	0.15 ± 0.05	0.20 ± 0.04	0.13 ± 0.04
EO (×10^3^/µL)	0.01 ± 0.01	0.00 ± 0.00	0.02 ± 0.01	0.02 ± 0.01	0.01 ± 0.01
BASO (×10^3^/µL)	0.00 ± 0.00	0.00 ± 0.00	0.00 ± 0.00	0.00 ± 0.00	0.00 ± 0.00

Results are expressed as mean ± S.E.M., *n* = 10, *n* = 5 (satellite group). * *p* < 0.05 compared with the control group. WBC, white blood cell; NEU, neutrophil; LYMP, lymphocyte; MONO, monocyte; EO, eosinophil; BASO, basophil.

**Table 7 biology-13-01005-t007:** Blood chemistry parameters of female rats in the chronic oral toxicity study of Maha Pigut Triphala.

Parameters	Control	Maha Pigut Triphala (mg/kg)			
		600	1200	2400	2400 (Satellite)
Glucose (mg/dL)	168.10 ± 7.69	164.90 ± 7.80	155.60 ± 18.45	154.00 ± 4.36	169.80 ± 10.24
BUN (mg/dL)	21.58 ± 0.66	21.93 ± 0.35	25.17 ± 3.27	23.00 ± 0.68	20.14 ± 1.07
Creatinine (mg/dL)	0.70 ± 0.02	0.73 ± 0.02	0.74 ± 0.02	0.72 ± 0.02	0.66 ± 0.01
Total protein (g/dL)	6.47 ± 0.10	6.52 ± 0.10	6.27 ± 0.18	6.27 ± 0.09	6.22 ± 0.09
Albumin (g/dL)	3.38 ± 0.04	3.36 ± 0.05	3.25 ± 0.08	3.29 ± 0.05	3.37 ± 0.04
Total bilirubin (mg/dL)	0.09 ± 0.00	0.09 ± 0.00	0.10 ± 0.01	0.09 ± 0.00	0.16 ± 0.02 *
Direct bilirubin (mg/dL)	0.12 ± 0.01	0.13 ± 0.02	0.15 ± 0.02	0.11 ± 0.02	0.10 ± 0.01
SGOT (U/L)	232.60 ± 25.99	181.80 ± 24.26	185.20 ± 17.43	162.70 ± 18.24 *	240.90 ± 33.25
SGPT (U/L)	63.80 ± 5.59	52.20 ± 5.50	55.60 ± 6.32	52.20 ± 6.77	77.90 ± 10.29
ALP (U/L)	43.60 ± 8.55	27.20 ± 2.41	63.30 ± 23.62	49.80 ± 10.52	31.40 ± 3.79

Results are expressed as mean ± S.E.M., *n* = 10, *n* = 5 (satellite group). * *p* < 0.05 compared with the control group. BUN, blood urea nitrogen; SGOT, serum glutamic oxaloacetate transaminase; SGPT, serum glutamate pyruvate transaminase; ALP, alkaline phosphatase.

**Table 8 biology-13-01005-t008:** Blood chemistry parameters of male rats in the chronic oral toxicity study of Maha Pigut Triphala.

Parameters	Control	Maha Pigut Triphala (mg/kg)			
		600	1200	2400	2400 (Satellite)
Glucose (mg/dL)	240.20 ± 18.98	214.60 ± 15.58	165.10 ± 14.97 *	231.70 ± 18.13	212.10 ± 11.28
BUN (mg/dL)	24.42 ± 0.95	24.51 ± 1.32	23.48 ± 0.59	23.34 ± 1.27	24.10 ± 0.75
Creatinine (mg/dL)	0.70 ± 0.03	0.68 ± 0.01	0.70 ± 0.02	0.67 ± 0.01	0.66 ± 0.02
Total protein (g/dL)	6.36 ± 0.29	6.16 ± 0.09	6.01 ± 0.06	6.17 ± 0.09	5.91 ± 0.07 *
Albumin (g/dL)	3.10 ± 50.07	3.07 ± 0.03	3.08 ± 0.03	2.99 ± 0.04 *	3.07 ± 0.03
Total bilirubin (mg/dL)	0.08 ± 0.01	0.07 ± 0.00	0.08 ± 0.00	0.07 ± 0.01	0.15 ± 0.02 *
Direct bilirubin (mg/dL)	0.10 ± 0.01	0.09 ± 0.00	0.11 ± 0.01	0.09 ± 0.00	0.10 ± 0.01
SGOT (U/L)	234.80 ± 27.18	215.10 ± 11.46	189.10 ± 11.84	203.80 ± 21.81	208.90 ± 13.98
SGPT (U/L)	79.70 ± 11.59	70.40 ± 6.51	62.80 ± 4.39	79.20 ± 8.50	71.90 ± 10.76
ALP (U/L)	85.30 ± 9.63	70.90 ± 7.14	62.80 ± 5.02	93.30 ± 21.47	53.50 ± 2.57

Results are expressed as mean ± S.E.M., *n* = 10, *n* = 5 (satellite group). * *p* < 0.05 compared with the control group. BUN, blood urea nitrogen; SGOT, serum glutamic oxaloacetate transaminase; SGPT, serum glutamate pyruvate transaminase; ALP, alkaline phosphatase.

## Data Availability

The original contributions presented in the study are included in the article/Appendix A, and further inquiries can be directed to the corresponding author.

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
