# Peer review of "Safety Evaluation for Acute and Chronic Oral Toxicity of Maha Pigut Triphala Contains Three Medicinal Fruits in Sprague-Dawley Rats"

_biology, 2024, doi:10.3390/biology13121005_

Round 1

Reviewer 1 Report

Comments and Suggestions for Authors

The authors assessed the safety of Maha Pigut Triphala herbal formula containing Terminalia bellirica, Terminalia chebula, and Emblica officinalis or Phyllanthus emblica extracts in the ratio of 2:1:3 employing Sprague-Dawley rats under acute and chronic oral regiments. The findings clearly show that Maha Pigut Triphala, in a ratio of 2:1:3, causes neither acute nor chronic oral toxicity in Sprague-Dawley rats, suggesting the safety of this herbal formula in this animal model. The study is well-designed, and the manuscript is well-written; thus, I recommend its publication with no further editing and revision. 

Reviewer 2 Report

Comments and Suggestions for Authors

The manuscript is written well. Please note a few minor edits I recommended in the document. 

Reviewer 3 Report

Comments and Suggestions for Authors

This study aimed to investigate the acute and chronic oral toxicity of Maha Pigut Triphala in the ratio of 2:1:3 in Sprague-Dawley rats. In general, this is an interesting topic but there are some shortcomings that have to be adequately addressed before the manuscript can be considered for publication.

My comments are as follows:

Major comments

1-The important point here is that the authors conducted previous research that dealt with the acute and chronic toxicity of Triphala, a formula called Triphala recipe, and proved that it is safe and non-toxic to rats (Arpornchayanon et al., 2022). Here, the authors studied the acute and chronic toxicity of the same plant, but with different concentrations of another formula, called Maha Pigut Triphala, which makes the importance of this research moderate.

2- The second observation is that the authors downplay the importance of some indicators that indicate chronic toxicity of the studied extract, such as enlargement of some organs such as the liver, lung, and kidney in males, as well as a significant increase in white blood cells and total bilirubin in the blood, which the authors consider to be an insignificant increase. This is inaccurate and needs to be re-discussed in a scientific manner.

Reference

1-Arpornchayanon, W., Subhawa, S., Jaijoy, K., Lertprasertsuk, N., Soonthornchareonnon, N. and Sireeratawong, S., 2022. Safety of the oral triphala recipe from acute and chronic toxicity tests in sprague-dawley rats. Toxics, 10(9), p.514.

Specific comments:

Methods

1-The doses used for Maha Pigut Triphala need to be determined in how they are chosen.

2-Some of the ethics for dealing with animals must be determined. How rats were anesthetized and what method of euthanasia was used here.

3-How was blood obtained from animals after death? Based on what the authors mentioned, it is assumed that the opposite is true. It should be mentioned how the blood samples were obtained.

Results

1-in the acute oral toxicity. Where are the changes in male weights? Why did the authors neglect them here?

2-The kidney weight for males in Table 2 needs to be corrected.

3-Why was the relative weight not calculated?

4-In Figures 5 and 6, the lung and liver sections have pathological changes and need to be changed. For the liver, the sections must be complete and not part of it. Also, for the kidney sections, it must contain Bowman’s capsule.

5-Regarding the authors’ interpretation of the Hematology results in the chronic study, it must be reinterpreted, separating the results for females from males, because there is a decrease in white blood cells in females, (Table 5), and an increase in white blood cells in males (Table 6).

6-In Table 8, the SGOT result was 234.80 ± 27.18 (U/L) for control rats, but that result is very high for the normal range of the reference values ​​for Sprague-Dawley rats, which is 59–139 (U/L) (He et al., 2017). What is your explanation and what method was used to estimate this enzyme?

Reference

1-He, Q., Su, G., Liu, K., Zhang, F., Jiang, Y., Gao, J., Liu, L., Jiang, Z., Jin, M. and Xie, H., 2017. Sex-specific reference intervals of hematologic and biochemical analytes in Sprague-Dawley rats using the nonparametric rank percentile method. PloS one, 12(12), p.e0189837.

Discussion

1-Discussion needs carful proofreading and copy-editing.

2- The authors’ interpretation of the results of WBC counts NEU, and LYMP count in males is illogical because the significant increase is large and its interpretation is the presence of chronic inflammation.

3-Also, a significant increase in total bilirubin in males and females is a significant increase indicating toxicity to the liver. Please explain it because it is accompanied by an enlargement of the weight of the liver and changes in histological sections.

Comments on the Quality of English Language

Discussion needs careful proofreading and copy-editing.
